

# Music emotion representation based on non-negative matrix factorization algorithm and user label information

Yuan Tian

School of Arts, Zhengzhou Technology and Business University, Zhengzhou, Henan, China

## ABSTRACT

Music emotion representation learning forms the foundation of user emotion recognition, addressing the challenges posed by the vast volume of digital music data and the scarcity of emotion annotation data. This article introduces a novel music emotion representation model, leveraging the nonnegative matrix factorization algorithm (NMF) to derive emotional embeddings of music by utilizing user-generated listening lists and emotional labels. This approach facilitates emotion recognition by positioning music within the emotional space. Furthermore, a dedicated music emotion recognition algorithm is formulated, alongside the proposal of a user emotion recognition model, which employs similarity-weighted calculations to obtain user emotion representations. Experimental findings demonstrate the method's convergence after a mere 400 iterations, yielding a remarkable 47.62% increase in F1 value across all emotion classes. In practical testing scenarios, the comprehensive accuracy rate of user emotion recognition attains an impressive 52.7%, effectively discerning emotions within seven emotion categories and accurately identifying users' emotional states.

## INTRODUCTION

Music serves as a powerful medium for expressing and evoking human emotions. As our understanding of user emotions continues to evolve, it has become imperative to develop more sophisticated and personalized emotion identification technology. However, since emotions are inherently subjective, many users refrain from expressing their emotions on social media, struggle to articulate their feelings accurately, or may not even be fully aware of their emotional states. Consequently, directly obtaining users' emotions presents a significant challenge. In response to this challenge, researchers have explored the notion that music can effectively convey and elicit emotions. Thus, organizing and retrieving music based on its emotional attributes has emerged as an objective approach. However, compared to other music concept recognition tasks, emotion recognition in music remains in its nascent stages. This is partly because the expression of music is subjective and challenging to quantify. Additionally, music emotion transcends mere audio data and cannot be comprehensively recognized solely based on audio features.

In the realm of music information retrieval, researchers have experimented with diverse music features to classify music emotions. Some studies have combined audio

Corresponding author
Yuan Tian, 3000007919@ztbu.edu.cn

and text features (*Houjeij et al., 2012*), while others have focused solely on audio or text. With the advent of machine learning technology, an increasing number of researchers employ methods like naive Bayes, SVM (*Dang & Shirai, 2009*) for classification, and KNN (*Furuya, Huang & Kawagoe, 2014*) and ANNOVA (*Choi, Lee & Downie, 2014*) for clustering, to tackle music emotion recognition challenges.

Currently, music emotion recognition employs feature selection strategies that typically involve selecting a set of frequency domain and time domain features based on researchers' experience. Alternatively, feature selection algorithms may be utilized to identify features from a broader range of frequency and time domain sets (*Dan, 2020*). However, using any single feature in isolation fails to yield optimal results, and existing feature engineering methods suffer from significant limitations and uncertainty. On music social platforms, while emotional tagging of song lists is easily obtainable, directly obtaining emotional information for individual music pieces remains challenging because users do not explicitly annotate the emotion of each song. Nevertheless, it is important to recognize that a song list essentially comprises a collection of individual music tracks. The inherent relationship between music and song lists provides an opportunity to calculate the emotion of individual music pieces indirectly.

In essence, the emotional attributes present in the song list can be inferred to characterize the emotions associated with each individual music track. This indirect approach enables researchers to estimate the emotional content of music without requiring explicit emotional annotations for each individual song, thus circumventing the direct annotation challenge and opening new possibilities for advancing music emotion recognition methodologies.

Matrix decomposition techniques can effectively reduce the dimensionality of the data. In the context of music emotion representation and user emotion recognition, this can be particularly valuable, as it helps capture the most relevant features of the data while eliminating noise or irrelevant information. Matrix decomposition can reveal latent features in the data that are not directly observable in the original representation. In music emotion analysis, these latent features could correspond to underlying patterns related to emotions. By extracting these features, the algorithm may achieve a more concise and expressive representation of emotional content in music. Therefore, the main goal of this article is to learn the representation of music in emotional space with the help of nonnegative matrix factorization (NMF) integrating external information, and based on user's listening list data and music emotion recognition architecture, a new user emotion recognition algorithm is designed, which can effectively identify the user's emotion in a short period of time.

## RELATED WORKS

### Emotional representation of music

At first, music emotion classification is based on audio features, but the effect of music emotion recognition only based on audio features is not ideal. With the development of multimodal information fusion technology, more and more scholars use the idea of

multimodality to retrieve music information. "Multimodal fusion" is the combination of different information sources and helps the system to complete specific tasks through the collaborative use of information provided by multiple modes (*Katsaggelos, Bahaadini & Molina, 2015*). Researchers have shown that the fusion of audio and lyrics can improve classification accuracy (*Laurier, Grivolla & Herrera, 2008*; *Yang et al., 2008*; *Yang & Lee, 2004*). *Jamdar et al. (2015)* and *Wang et al. (2020)* used keyword discovery technology in the process of lyrics processing, and extracted melody MFCC and pitch features. Experimental results show that the combination of the two features can significantly improve the performance. *Laurier, Grivolla & Herrera (2008)* used language model differences to select words, moreover, the lyrics were characterized by bag of words and combined with acoustic features to improve the accuracy of emotion classification; *Yang et al. (2008)* transformed lyrics into 182 psychological categories, and the fusion features improved the classification effect.

## User emotion recognition

In the field of music information retrieval, more and more researchers combine music emotion with user emotion to carry out research. Music is not only emotional in itself, but also evokes the emotions of the audience (*Qianwen, 2022*). The ability of music to induce emotions ensures its prominent position in human culture and daily life. Many studies have proposed a variety of methods to classify and identify users' emotional states. *Kiritchenko, Zhu & Mohammad (2014)* constructed an emotion analysis system to identify emotions in informal texts such as tweets and messages, and achieved excellent results on SMS test set and film review corpus; *Godbole, Srinivasaiah & Skiena (2007)* tested public sentiment for news, track the evolution of public sentiment index and support rate for the president; *Saari & Eerola (2014)* uses social tags to identify music audience emotions. In addition, many researchers use dictionary based methods to carry out research. For example, *Wilson, Wiebe & Hoffmann (2005)* constructed the subjective vocabulary by questioning the respondents, and finally got about 8,000 emotional tags; The NRC emotion lexicon constructed by *Mohammad & Turney (2010)* through Mechanical Turk has about 14,000 words of emotion and tag. Machine learning can effectively solve the problem of emotion dictionary, but it cannot solve the problem of unknown words. *Lixing, Ming & Maosong (2012)* proposed a method for sentiment classification of Twitter using word vector and realized effective supervision of the emotional polarity of text with the help of three neural networks.

## MUSIC EMOTION REPRESENTATION MODEL BASED ON NMF

Non-negative matrix decomposition (NMF) combines cryptic meaning and machine learning characteristics, which can dig deeper connections between users and items. Therefore, the prediction accuracy is relatively high, and the prediction accuracy is higher than neighborhood based collaborative filtering and content-based recommendation algorithm. The model can be trained by stochastic gradient descent method and alternating least square method. At the same time, matrix decomposition has low time and space
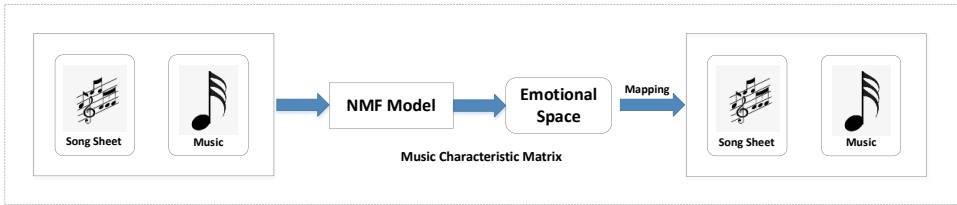

**Figure 1   NMF music emotion model.**

complexity. The mapping of high-dimensional matrix into two low-dimensional matrices saves storage space, and the training process is time-consuming, but can be completed offline. Score prediction is generally calculated online, directly using the parameters obtained from offline training, which can be recommended in real time.

## NMF model

The NMF technology is specifically explained. The main problem of the model is transformed into the structure shown in Fig. 1: construct a music feature matrix, use NMF to map the song list and music to the emotional space at the same time, and obtain the representation or distribution of the song list and music in the emotional space. The co-occurrence information is represented as a two-dimensional matrix. This type of matrix is a grid-like structure with rows and columns, forming a table. In this context, the rows correspond to different song lists, and the columns correspond to individual musical pieces. The elements of the matrix represent the co-occurrence strength or count of each musical piece in each song list. For example, if element (i, j) of the matrix is 3, it means that the j-th musical piece appears three times in the i-th song list.

By using this two-dimensional matrix to represent the co-occurrence information, researchers, data analysts, or music enthusiasts can analyze patterns and relationships between different song lists and musical pieces. It allows them to discover which songs are commonly associated with specific playlists or which playlists share similar songs, thus aiding in music recommendation systems, playlist generation, and other music-related data analyses. The co-occurrence information of playlist and music is represented as a two-dimensional matrix, in which each row represents a playlist and each column represents a piece of music. The value in the matrix is 0 or 1, indicating whether the corresponding music appears in the playlist (1 for the occurrence, 0 for the absence). Due to the large number of music in the list, there will be problems such as excessive resource consumption and time-consuming in data calculation. The form of matrix X should be preprocessed as follows:

$$X = \begin{pmatrix} a_1 1 & \cdots & a_{1n} \\ \vdots & \ddots & \vdots \\ a_{m1} & \cdots & a_{mn} \end{pmatrix} \tag{1}$$

where m represents the singular number of songs and n represents the number of music

A two non-negative factorization matrices of the X should be arranged to replace U and V, which make the original matrix X as close as possible to the factorized result $UV^T$. as shown in Formula (2):

$$O = \left\| X - UV^T \right\|_F^2$$
$$\text{s.t } U \geq 0, V \geq 0 \tag{2}$$

## Emotional labels

In the context of music emotion representation, the song sheet or music emotional label information which is labeled by experts is included in the matrix decomposition as external information. Assuming that the following objective functions should be minimized as follows:

$$\left\| G_v (V - V_0) \right\|_F^2. \tag{3}$$

Among them, $V_0$ represents music-emotion label matrix, $G_v$ is the diagonal indicator matrix representing the musical level. $G_v(i, i) = 1$ indicates that the I track contains an emotional indicator, and $G_v(i, i) = 0$ if it does not.

## Emotional representation

The quintessence of music feeling acknowledgment is the portrayal of music in the profound space. Let the emotional distribution of music i be $V_i$, then music i can be expressed in the k-dimensional emotional space as:

$$V_i = (V_{i1}, V_{i2}, \ldots, V_{ik}). \tag{4}$$

The $V_i$ corresponding to music i is normalized, and the emotional probability distribution $s_i^*$ corresponding to music i is obtained after normalization. The calculation formula is as follows:

$$s_i^* = \left( \frac{V_{ij}}{\sum_{j=1}^{k} V_{ij}} \right) \tag{5}$$

where k represents the dimension of emotional space.

For each piece of music, the highest emotion category is taken. The calculation formula is as follows:

$$e_i^* \leftarrow \arg \max \left( s_i^*. \right) \tag{6}$$

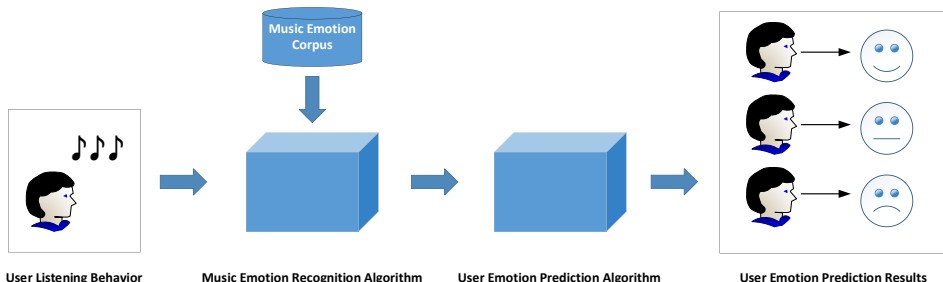

**Figure 2  Framework of user emotion recognition.**

## Emotional computing

Through the above calculation, the distance relationship between music and music is obtained. The first k music most similar to music i is regarded as the neighbor of music i, which is represented by N(i). Assume that the emotion representation vector of unknown music i is $s_i$, and the emotion distribution vector of the j-th neighbor of music i is $s_j^*$. Use the average value of the emotion vector of neighbor N(i) of music i to represent the emotion of unknown music i, then the calculation formula of the emotion representation vector of music i is as follows:

$$s_i^* = \frac{\sum_{i=1}^{N(i)} s_j^*}{\sum_{j=1}^{N(i)} s_j^*}.$$ 

(7)

Finally, the emotional category of music with the maximum value is selected as the result of emotional classification of music, and the emotional category to which music i ultimately belongs is s $e_i^*$, which can be calculated as follows:

$$e_i^* \leftarrow \arg\ max\left(s_i^*\right).$$ 

(8)

# USER EMOTION RECOGNITION MODEL

## Overall framework

The music that users have listened to forms the music list. In order to realize the recognition of users' immediate emotions, the emotion of each music in the music list should be identified first. Combined with the user's historical listening behavior, an emotion recognition algorithm is designed to calculate the user's emotional perception when listening to songs. Finally, the emotion of users listening to songs is classified into "cure, relaxing, romantic, nostalgic, excited, lonely and quiet". Figure 2 shows the framework of user emotion recognition.

The proposed framework encompasses three main components: Melody Compilation, Sentiment Elicitation, and Affective Computation. Firstly, in the Melody Compilation

phase, it is essential to acquire historical user data from their music-listening experiences. Generally, users create a song compilation, serving as a curated list of songs. Secondly, in the Sentiment Elicitation phase, a specialized algorithm for music sentiment extraction is designed to capture all emotions embedded in the song compilation, whether overt or latent. Lastly, during the Affective Computation stage, the acquired music sentiments are utilized to compute the user's emotional representation based on their music compilation. This computation yields distinct emotional categories, thereby accomplishing the objective of emotion recognition. Ultimately, the outcomes of user emotion recognition can be applied to various specific scenarios.

## Algorithm design

First, the user emotion recognition is transformed into a mathematical problem. Let the emotion distribution vector of the i-th user be $u_i^*$, and there are k classes user emotion at present, and the emotion of user i can be expressed as:

$$u_i = \left( u_i^1, \ldots, u_i^j, \ldots, u_i^k \right) \tag{9}$$

Based on the user's listening list, the number of times the user listens to songs is taken as the weight, and the weighted average value of the emotional vector of music in the historical listening list represents the user's emotions, and the user's emotional distribution is finally obtained. The corresponding weight of song j in the listening list is set as $w_j$, and the weight calculation method is as follows: If music ranks first, that is, the number of times users listen to songs is the largest, then the weight of music is 100; While if the music is not ranked first, the corresponding weight = the number of times the music is listened to/the times music ranked first is listened to $\times 100$

Assuming that the distribution vector of music emotion is $s_j^*$, and the emotion of music can be expressed as:

$$s_j = \left( s_j^1, \ldots, s_j^k \right) \tag{10}$$

The calculation formula of user i's emotional representation is as follows:

$$u_i^* = \frac{\sum_{j=1}^{k} w_j * s_j^*}{\sum_{j=1}^{k} w_j * s_j^*} \tag{11}$$

After obtaining the emotional distribution vector of the user, the emotional dimension of the maximum value is taken as the final emotion of the user. The final emotional class of user i is expressed as follows:

$$e_i^* \leftarrow \arg max \left( u_i^* \right). \tag{12}$$

**Table 1   Explanation of variables in F1.**

|  | The classifier is identified as i | Classifier recognition is not i |
|---|---|---|
| Manually marked as i | TP | FN |
| Manual marking is not i | FP | TN |

## EXPERIMENTAL RESULTS

### Data set

The dataset utilized in this research is derived from the "Music Emotion Dataset". It comprises user-generated song lists, along with corresponding label data assigned by users. The dataset encompasses a total of 13,565 song lists, encompassing more than 490,000 individual songs. Each song list is associated with five distinct categories of tags, namely language, style, scene, emotion, and theme.

For the experimentation, a subset of 4,200 music pieces is selected as a sample. Within this sample, 4,927 songs are designated as the feature space, with each song represented as a music vector that characterizes its distribution across the attributes of the song list.

The experimental design entails employing a five-fold cross-validation approach. This methodology involves dividing the dataset into five subsets, and for each iteration, four subsets are used as the training set, while the remaining subset serves as the test set. This process is repeated five times, ensuring that each subset is utilized as the test set once, thereby facilitating robust evaluation and validation of the model's performance. Cross-validation is valuable when tuning hyperparameters of the model. It allows us to iteratively evaluate the model's performance with different hyperparameter settings and choose the best combination that generalizes well across various subsets of the data. data distributions may change over time due to various factors (data drift). Cross-validation can help identify how well the model generalizes across different data distributions, providing insights into the model's robustness.

In the part of user emotion recognition, the listening records of 24,537 users for nearly a week since October 6, 2021 were crawled from "Netease cloud music", including user ID, music ID and times of listening, which is the corresponding weight.

### Evaluation index

F1 is used to measure the accuracy of the model in the multi-classification task, and the value range is 0-1. Before calculating F1, it is necessary to map musical emotion to the emotion class with the maximum corresponding value from the musical emotion distribution in the music-emotion matrix V, and then calculate. The higher the F1 value is, the better the training effect is. In this experiment, each group of tests were carried out three times, and the average value was taken as the results for discussion. The calculation of F1 is expressed as Formula (13) to Formula (15), and the meaning of each variable is shown in Table 1.

$$\text{Precision}_{\text{mi}} = \frac{\sum_{i=1}^{n} TP_i}{\sum_{i=1}^{n} TP_i + \sum_{i=1}^{n} FP_i} \tag{13}$$

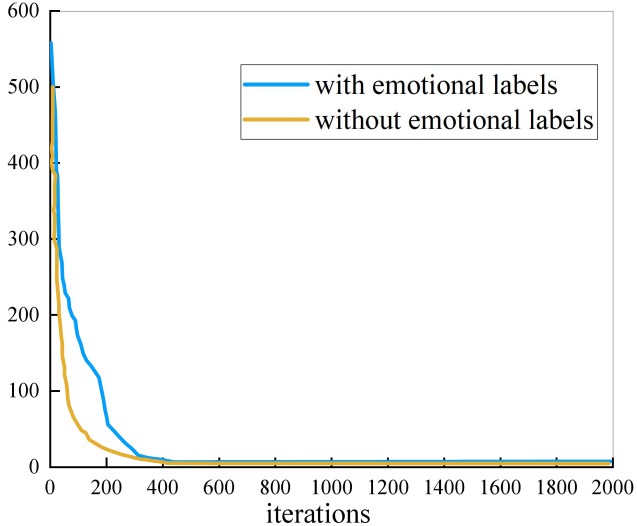

**Figure 3** The evolutionary process of NMF.

$$\text{Recall}_{\text{mi}} = \frac{\sum_{i=1}^{n} \text{TP}_i}{\sum_{i=1}^{n} \text{TP}_i + \sum_{i=1}^{n} \text{FN}_i} \qquad (14)$$

$$\text{F1} = \frac{2 \star \text{Precision} \star \text{Recall}}{\text{Precision} + \text{Recal}}. \qquad (15)$$

## Validity of the algorithms

Figure 3 shows the evolution process of NMF. As can be seen from the figure, the convergence speed of the algorithm without emotional labels is always faster than that of algorithm with emotional labels. The convergence of the two methods is similar, reaching convergence within 400 iterations.

Figure 4 shows the *F* value of the algorithm on each emotion class. Adding emotional tags had a significant impact on the emotion of "Excitement", and its F1 increased by 47.62%. "Nostalgia" is generally the best emotional class, while the "quiet" is generally poor, which may be due to the fact that compared with other emotional categories, the performance of "Nostalgia" is generally the best, Listening groups generally have similar cognition on whether music is "Nostalgia", while "Quiet" and "Cure", "relaxing" and other emotional categories overlap, which is not easy to distinguish.

In order to further verify the effectiveness of the matrix decomposition method in the music domain data set generated by users, the method in this article is compared with the other five methods: K"-means, support vector machine (SVM), logistic regression (LR), AdaBoost, decision tree (DT). The results are shown in Fig. 5.

The comprehensive analysis of the experimental results reveals intriguing insights into the performance of various models. The non-negative matrix factorization (NMF)
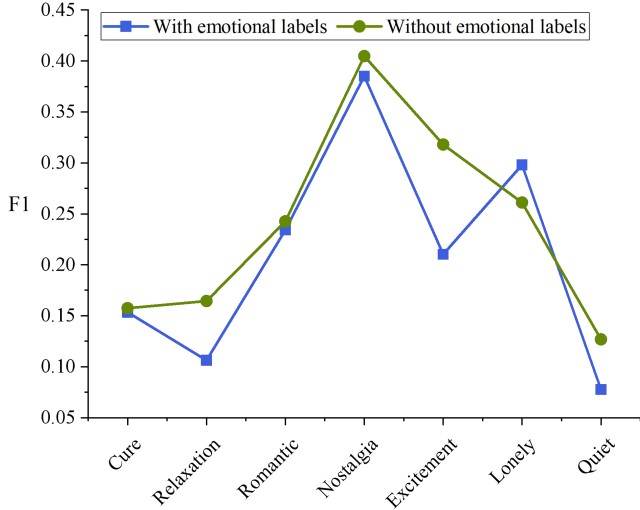

**Figure 4   F1 values of different emotional classes.**

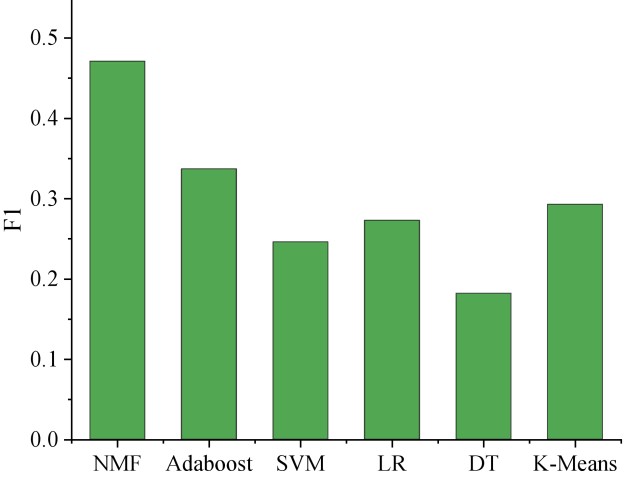

**Figure 5   Comparison of different learning methods.**

technique, when employed with emotional labels, showcases the best overall performance, surpassing AdaBoost, the second-ranking model, by a remarkable margin of approximately 12.5%. This noteworthy difference indicates the superiority of nonnegative matrix factorization (NMF) in capturing underlying patterns and relationships among the emotional features, leading to more accurate classification results.

Notably, k-means exhibits comparable outcomes to logistic regression (LR). The k-means algorithm partitions data into clusters, effectively separating instances with similar features. This behavior aligns with LR's capacity to find decision boundaries in a linear manner, hence yielding similar results. However, NMF provides an added

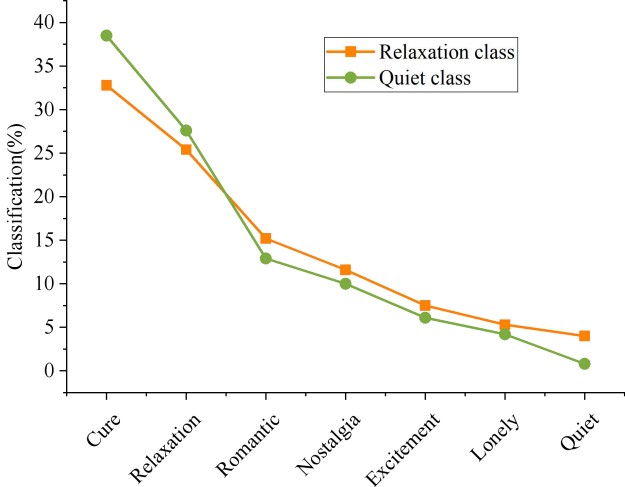

**Figure 6** Classification of music emotion.

advantage by decomposing the original matrix into two non-negative matrices that approximately approximate the initial data, thereby mapping features into a hidden space simultaneously. This dual mapping contributes to enhanced classification efficacy.

On the other hand, the decision tree model's performance falls short, displaying inferior results compared to other models. Decision trees are prone to overfitting and may struggle to generalize well on complex datasets. Consequently, this negatively impacts the model's recognition effectiveness.

Overall, while some models perform similarly, it is evident that NMF with emotional labels significantly outperforms the rest. Its ability to extract meaningful representations from the data, combined with a dual mapping approach, highlights its potential for various applications, such as sentiment analysis, emotion recognition in text or speech, and user behavior prediction. Further research could explore parameter tuning and feature engineering to optimize the models further, ultimately advancing the field of emotion-based machine learning and data analysis.

## User emotion recognition
### Music classification

In order to better identify users' emotions, it is necessary to classify music accurately. The algorithm in Fig. 4 performs poorly in the two clesses of "Relaxation" and "Quiet". Therefore, the final emotion classification of music that should belong to these two classes is calculated, and the results are shown in Fig. 6.

More than half of the music that should have belonged to the class of "Relaxation" was wrongly assigned to the category of "Cure" and "Nostalgia", while less was wrongly assigned to the category of "Lonely". This phenomenon is consistent with cognition. The two class of "Relaxation" and "healing" do overlap. The concert of "Cure" brings people the feeling of "Relaxation". Similarly, relaxing music can also make people feel "Cure". Most (40.67%) of the music that should have belonged to the class "Quiet"

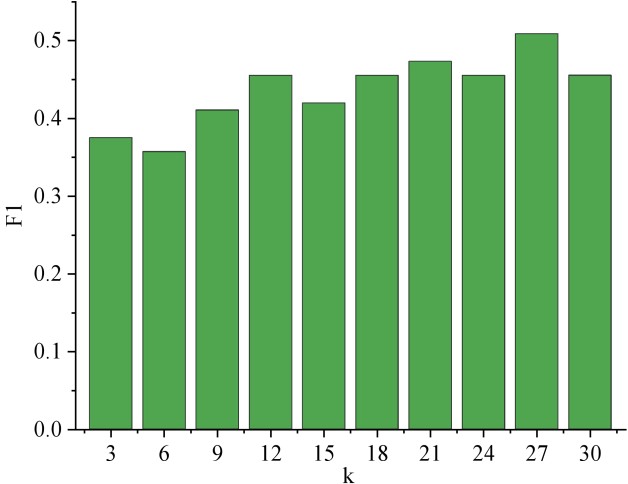

**Figure 7  Variation Trend of F1 with k.**

were wrongly assigned to the class "Cure", followed by the class "Nostalgia". Combined with the practical experience in daily life, most of the music under the "Quiet" is pure music, or relatively soft and soothing music, and these music may also have the function of "Cure", which can regulate people's mood and emotion and make people calm down. Therefore, this phenomenon is reasonable and conforms to common sense cognition.

To sum up, due to the limitation of data, the reason for this phenomenon is that the limitation of F1. Music contains only one emotion tag, that is, the emotion class with the highest value, while the emotion of music does not necessarily belong to a single class. The above results are consistent with the actual situation, so the reliability of the algorithm in this article is verified.

### Analysis of user emotion recognition

When identifying user emotions, the number of nearest neighbor music k of unknown music will affect the result of music emotion recognition, and then affect the result of user emotion recognition. Therefore, parameter experiments need to be carried out around k. Figure 7 shows the trend of F1 with parameter k when identifying user emotions.

The analysis of Fig. 7 reveals a significant trend concerning the performance of F1 with varying values of k. As k increases, F1 demonstrates a consistent improvement until reaching its peak at $k = 27$, with a remarkable score of 0.509. This suggests that the parameter k profoundly impacts the overall effectiveness of the system. Among the 119 users, a substantial portion of 59 users achieved accurate emotion recognition, resulting in an impressive accuracy rate of 52.7%. This finding underscores the system's reliability in detecting emotions, positively impacting user experiences and applications in various fields such as sentiment analysis, user feedback processing, and emotional well-being

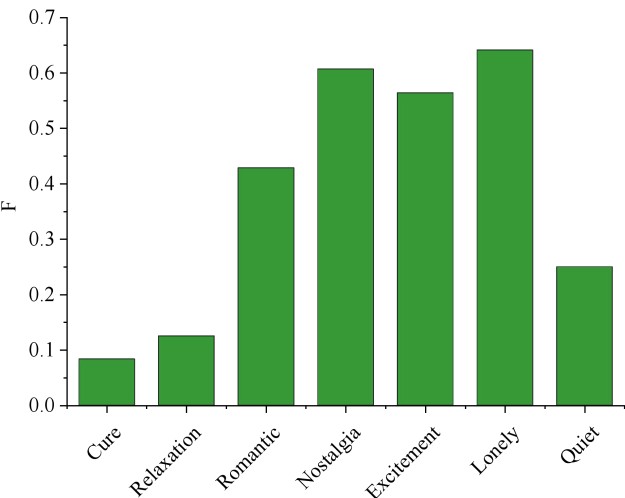

**Figure 8  Result of user emotion recognition.**

assessments. Further studies could explore optimal k-values and refine the system for even better performance. The result of user emotion recognition is shown in Fig. 8.

The algorithm performs well in the four classes of "Romance", "Nostalgia", "Excitement" and "Lonely". The algorithm performs best in the class of "Lonely", with the optimal F value up to 0.6415, the highest F value in the category of "Nostalgia" is 0.6071, and the highest F value in the category of "Excitement" is 0.5641. In the "Romantic" category, $F$ value is 0.4286; The highest $F$ value in the "Cured" category was 0.2069, the highest $F$ value in the "Relaxed" category was 0.125 and the highest $F$ value in the "Quiet" category was 0.25. Similar to the previous content, it shows that this is due to the similarity of the three classes, high degree of coincidence and close emotion. For example, "Cure" and "Quiet" music can also make people feel relaxed. Most relaxing music is gentle and soothing, so it is difficult to distinguish them. The other four categories have relatively obvious characteristics, such as "Excitement" music usually has a faster rhythm, while "Lonely" music usually have slow rhythm and sad words. In short, the overall performance of the algorithm in this article is good, which can effectively distinguish emotions under seven emotion categories and identify users' emotions.

## APPLICATION SCENARIOS

Emotional analysis is divided into three stages: emotional expression stage, emotional perception stage and emotional induction stage. The emotional communication of users in the process of enjoying music can also be divided into three stages: firstly, the emotion of music is expressed through audio and lyrics, which is the emotion that the performer or composer wants to express; then, when the user hears the music, the emotion in music is perceived by users; finally, the user's own emotion is aroused, that is, the user's emotional response when listening to music. The user's emotional response is

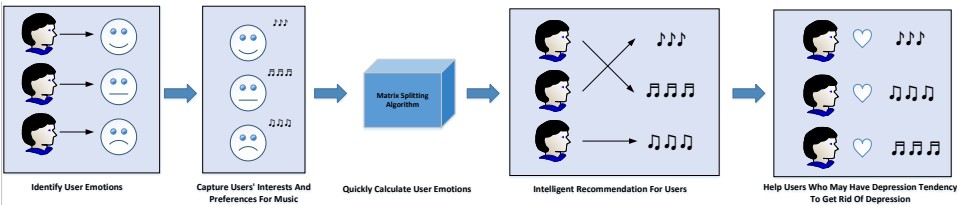

**Figure 9 Application scenario.**

not necessarily consistent with the perceived emotion, which may be related to the user's own experience and mood at that time.

This article aims to study the emotion of users when they hear music, and a set of algorithm is designed to identify user emotion, which has important practical application value. The specific application scenario is shown in Fig. 9.

(1) Music platform:

The optimization of recommendations on the music platform can be achieved through the discernment of user emotions, as emotion represents a pivotal dimension in the recommendation process. By classifying music based on emotions, it becomes feasible to accurately capture users' inclinations and preferences for music imbued with varying emotional content, thereby refining the recommendation system. Furthermore, the algorithm introduced in this article facilitates swift computation of user emotions, empowering the platform to monitor users' emotional states in real-time and grasp the latest emotional trends. By discerning users' moods, the platform can execute more intelligent marketing strategies. For instance, if a user's emotional level appears negative for an extended period, the platform can recommend content like relaxing books or travel packages to uplift their spirits.

(2) Depression treatment:

Additionally, the findings of this research hold practical significance in the domains of user psychology and social psychology. Psychological studies have substantiated that individuals afflicted with depression often gravitate towards low-energy and melancholic music due to the soothing impact of such compositions (*Wenwen, 2021*). For depression patients, communication challenges and hesitancy to seek help are common barriers; however, listening to music remains a habitual activity in their daily lives. By tracking users who frequently engage with low-energy and sad music, their emotions can be effectively identified, with special attention directed towards those exhibiting pronounced negative emotional tendencies (*Chandane et al., 2020*). In addition, a psychological counseling feature can be integrated, offering tailored music therapy programs to potential depression patients and aiding in the treatment of depression. Such proactive measures can provide timely support to users displaying depressive tendencies, helping them overcome this state and reclaim their well-being.

Moreover, matrix decomposition can easily integrate data from multiple modalities, such as combining audio features with lyrics or user-specific contextual information. This integration can lead to a richer and more comprehensive representation of emotions.

The matrix decomposition approach can be adapted to capture individual differences in emotion perception. By learning personalized representations for users, the algorithm can enhance emotion recognition accuracy and provide more tailored music recommendations.

## CONCLUSION

In this article, starting from the user's behavior of creating a song list and assigning a label to the song list, an NMF algorithm integrating emotional tags is proposed. Based on the massive social media data generated by users, the emotion of music is represented. The results show that this method can effectively reduce the cost of manual annotation, and the integration of external information can effectively improve the effect of music emotion recognition. In addition, a music emotion recognition algorithm is designed, which can directly obtain the corresponding emotion of music, or obtain the emotional representation of the music through similarity weighted calculation. Experiments show that the algorithm in this article performs well on the whole, and can effectively distinguish emotions under seven emotion categories and identify users' emotions, which can effectively identify the emotion contained in massive unknown music. In specific application scenarios, the rational use of the model is expected to help users with depression get rid of depression in time.

## ACKNOWLEDGEMENTS

We would like to thank all the anonymous reviewers who have suggested improvements to this article.

### Funding
This work received no funding.

### Competing Interests
The authors declare there are no competing interests.

### Author Contributions
- Yuan Tian conceived and designed the experiments, performed the experiments, analyzed the data, performed the computation work, prepared figures and/or tables, authored or reviewed drafts of the article, and approved the final draft.

### Data Availability
The code is available in the Supplemental Files.

The data is available in Zenodo: Chua, Phoebe, Makris, Dimos, Herremans, Dorien, Roig, Gemma, & Agres, Kat. (2022). Predicting emotion from music videos: exploring the relative contribution of visual and auditory information to affective responses. [Data set]. Zenodo. https://doi.org/10.5281/zenodo.7128177.

## Supplemental Information

Supplemental information for this article can be found online at http://dx.doi.org/10.7717/peerj-cs.1590#supplemental-information.

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
