# Peer review of "Music emotion representation based on non-negative matrix factorization algorithm and user label information"

_PeerJ Computer Science, doi:10.7717/peerj-cs.1590_

## Round 0.1 · original submission · Major Revisions

Dear authors,

Thank you for your submission. Your article has not been recommended for publication in its current form. However, we do encourage you to address the concerns and criticisms of the reviewers and resubmit your article once you have updated it accordingly.

Best wishes,

**Language Note:** The review process has identified that the English language must be improved. PeerJ can provide language editing services - please contact us at copyediting@peerj.com for pricing (be sure to provide your manuscript number and title). Alternatively, you should make your own arrangements to improve the language quality and provide details in your response letter. – PeerJ Staff

Reviewer 1 ·

Basic reporting

This paper introduces an innovative model for representing musical emotions. By capturing the essence of emotional experiences, the model effectively learns to represent music within the emotional domain. This is accomplished by integrating a refined algorithm, constrained non-negative matrix decomposition, with emotional label information derived from user-generated listening song lists. Consequently, the model successfully achieves the goal of emotion recognition. Furthermore, a novel algorithm for music emotion recognition is devised, along with the proposal of a user emotion recognition model. The latter method utilizes similarity weighting calculations to obtain a comprehensive representation of user emotions. While this paper exhibits elements of innovation, some modifications are necessary to enhance its overall quality.
1- The abstract of the article requires enhancement to achieve conciseness and comprehensiveness.
2- The reviewers concur that this study is of paramount importance, encompassing a considerable patient cohort. Nevertheless, the presence of grammatical errors in English undermines the paper's quality and hinders its readability.
3- Regarding the sentence, "The co-occurrence information of the song list and music is depicted as a two-dimensional matrix, wherein each row corresponds to a song list, and each column corresponds to a musical piece." Further elaboration is required to elucidate the relationship between "song list" and "music."

Experimental design

1- This paper employs a matrix decomposition algorithm for music emotion representation and user emotion recognition. It is essential to elucidate the merits of the matrix decomposition algorithm in contrast to alternative methodologies.
2- The author's choice of keywords necessitates additional refinement. Attention should be given to the inclusion of "depression" as an emotional label, as it leads to logical redundancy.

Validity of the findings

1- The manuscript ought to explicitly specify the cognitive model employed by alternative methods for comparative analysis. Furthermore, it is crucial to outline the advantages conferred by this innovative approach.
2- Regarding the sentence, "The co-occurrence information of the song list and music is depicted as a two-dimensional matrix, wherein each row corresponds to a song list, and each column corresponds to a musical piece." Further elaboration is required to elucidate the relationship between "song list" and "music."

Reviewer 2 ·

Basic reporting

In this paper, a matrix decomposition algorithm based on emotion tags is introduced to characterize music emotion based on massive user-generated social media data. The results show that this method can effectively reduce the cost of manual annotation, and the integration of external information can effectively improve the effect of music emotion recognition. In addition, a musical emotion recognition algorithm is designed, which can directly obtain the corresponding emotion of the music or obtain the emotional representation of the music through similarity weighting calculation.
However, there are several issues that require revision to ensure the acceptance of the manuscript. The common problems identified are as follows:

1. The author proposes a specific application scenario for this model, which holds significant meaning. However, it is suggested to describe the scenario separately for different stages.

2. The introduction's first part is overly verbose, with repeated statements. The author is recommended to modify it accordingly.

3. It is unclear whether the selected data from the digital music social platform includes user-labeled data and what the label categories are.

4. In general, there is a lack of explanation regarding the replication and statistical methods employed in the study.

5. Furthermore, an explanation of the rationale behind conducting these various experiments should be provided.

6. The description of the overall framework of the user emotion recognition model in section 4 should be more concise.

7. There are numerous semantic deficiencies in the article, such as the statement: "If the emotional representation learned by matrix decomposition is inconsistent with the emotional indication contained in the data, the loss function will generate a penalty, thus optimizing the direction of the next decomposition in matrix decomposition." It is unclear how this method further optimizes matrix decomposition.

8. The comparative analysis of the results obtained from different learning methods is relatively weak. The author should analyze the advantages of the results based on the characteristics of the adopted models.

9. Some statements lack smoothness, resulting in poor readability of the article. These areas require further modification.

Experimental design

In this paper, a matrix decomposition algorithm based on emotion tags is introduced to characterize music emotion based on massive user-generated social media data. The results show that this method can effectively reduce the cost of manual annotation, and the integration of external information can effectively improve the effect of music emotion recognition. In addition, a musical emotion recognition algorithm is designed, which can directly obtain the corresponding emotion of the music or obtain the emotional representation of the music through similarity weighting calculation.
However, there are several issues that require revision to ensure the acceptance of the manuscript. The common problems identified are as follows:

1. The author proposes a specific application scenario for this model, which holds significant meaning. However, it is suggested to describe the scenario separately for different stages.

2. The introduction's first part is overly verbose, with repeated statements. The author is recommended to modify it accordingly.

3. It is unclear whether the selected data from the digital music social platform includes user-labeled data and what the label categories are.

4. In general, there is a lack of explanation regarding the replication and statistical methods employed in the study.

5. Furthermore, an explanation of the rationale behind conducting these various experiments should be provided.

6. The description of the overall framework of the user emotion recognition model in section 4 should be more concise.

7. There are numerous semantic deficiencies in the article, such as the statement: "If the emotional representation learned by matrix decomposition is inconsistent with the emotional indication contained in the data, the loss function will generate a penalty, thus optimizing the direction of the next decomposition in matrix decomposition." It is unclear how this method further optimizes matrix decomposition.

8. The comparative analysis of the results obtained from different learning methods is relatively weak. The author should analyze the advantages of the results based on the characteristics of the adopted models.

9. Some statements lack smoothness, resulting in poor readability of the article. These areas require further modification.

Validity of the findings

In this paper, a matrix decomposition algorithm based on emotion tags is introduced to characterize music emotion based on massive user-generated social media data. The results show that this method can effectively reduce the cost of manual annotation, and the integration of external information can effectively improve the effect of music emotion recognition. In addition, a musical emotion recognition algorithm is designed, which can directly obtain the corresponding emotion of the music or obtain the emotional representation of the music through similarity weighting calculation.
However, there are several issues that require revision to ensure the acceptance of the manuscript. The common problems identified are as follows:

1. The author proposes a specific application scenario for this model, which holds significant meaning. However, it is suggested to describe the scenario separately for different stages.

2. The introduction's first part is overly verbose, with repeated statements. The author is recommended to modify it accordingly.

3. It is unclear whether the selected data from the digital music social platform includes user-labeled data and what the label categories are.

4. In general, there is a lack of explanation regarding the replication and statistical methods employed in the study.

5. Furthermore, an explanation of the rationale behind conducting these various experiments should be provided.

6. The description of the overall framework of the user emotion recognition model in section 4 should be more concise.

7. There are numerous semantic deficiencies in the article, such as the statement: "If the emotional representation learned by matrix decomposition is inconsistent with the emotional indication contained in the data, the loss function will generate a penalty, thus optimizing the direction of the next decomposition in matrix decomposition." It is unclear how this method further optimizes matrix decomposition.

8. The comparative analysis of the results obtained from different learning methods is relatively weak. The author should analyze the advantages of the results based on the characteristics of the adopted models.

9. Some statements lack smoothness, resulting in poor readability of the article. These areas require further modification.

Additional comments

pl. see above comments

---

## Round 0.2 · accepted · Accept

Dear authors,

Thank you for the revision. The paper seems to be improved in the opinion of the reviewers. The paper is now ready for publication.

Best wishes,


Reviewer 1 ·

Basic reporting

All changes have been incorporated in the updated manuscript.

Experimental design

All changes have been incorporated in the updated manuscript.

Validity of the findings

All changes have been incorporated in the updated manuscript.

Reviewer 2 ·

Basic reporting

This paper proposes a non-negative matrix factorisation algorithm for user-label information. The idea is good

Experimental design

The experimental design has been further improved in the revised version of the paper and therfore, i do not have any further concerns on it.

Validity of the findings

The study findings are satisfied and improved in revised version.

Additional comments

The paper is improved in light of my previous comments, therefore, i recommend it